# Preclinical Anticipation of On- and Off-Target Resistance Mechanisms to Anti-Cancer Drugs: A Systematic Review

**DOI:** 10.3390/ijms25020705

**Published:** 2024-01-05

**Authors:** Paulina J. Dziubańska-Kusibab, Ekaterina Nevedomskaya, Bernard Haendler

**Affiliations:** Research and Early Development Oncology, Pharmaceuticals, Bayer AG, Müllerstr. 178, 13353 Berlin, Germany; paulina.dziubanska-kusibab@bayer.com (P.J.D.-K.); ekaterina.nevedomskaya@bayer.com (E.N.)

**Keywords:** cancer treatment resistance, deep mutational scanning, CRISPRko, CRISPRi, CRISPRa, CRISPR base editing

## Abstract

The advent of targeted therapies has led to tremendous improvements in treatment options and their outcomes in the field of oncology. Yet, many cancers outsmart precision drugs by developing on-target or off-target resistance mechanisms. Gaining the ability to resist treatment is the rule rather than the exception in tumors, and it remains a major healthcare challenge to achieve long-lasting remission in most cancer patients. Here, we discuss emerging strategies that take advantage of innovative high-throughput screening technologies to anticipate on- and off-target resistance mechanisms before they occur in treated cancer patients. We divide the methods into non-systematic approaches, such as random mutagenesis or long-term drug treatment, and systematic approaches, relying on the clustered regularly interspaced short palindromic repeats (CRISPR) system, saturated mutagenesis, or computational methods. All these new developments, especially genome-wide CRISPR-based screening platforms, have significantly accelerated the processes for identification of the mechanisms responsible for cancer drug resistance and opened up new avenues for future treatments.

## 1. Introduction

The use of targeted molecular therapy and precision medicine has led to dramatic progress in the treatment of cancer, which, together with changes in lifestyle and earlier detection, translate into significantly improved outcomes [1]. This is particularly noticeable in tumors of the lung, colon, breast, and prostate, which belong to the most frequent cancer types worldwide and are predicted to represent 12.5%, 8%, 31% (in females), and 29% (in males) of new cancer cases in the United States in 2023, respectively, and now all have declining mortality rates ranging between 1.2% and 4% per year for the 2011–2020 time period [1]. According to another, worldwide survey from 2020, the most frequently detected tumor forms were also female breast cancer (11.7%), lung cancer (11.4%), colorectal cancer (10%), and prostate cancer (7.3%) [2]. Unfortunately, treatments leading to curing cancer are still exceedingly rare, with the gains in survival rates being the highest in hematopoietic and lymphoid tumors, mainly due to the development of tyrosine kinase inhibitors [1]. Concerning solid tumors, the advent of immune checkpoint inhibitors has led to major advances, especially in melanoma, with a 258% increase in patients living with metastatic disease between 1990 and 2018 [1]. In many cases, however, patients will ultimately relapse following these targeted therapies and invasive processes, leading to distant metastases, occur.

Currently, efforts to understand and overcome resistance take place mostly after loss of drug efficacy is observed in patients and when sufficient clinical samples are available for in-depth molecular characterization (Figure 1a). Collection and detailed analysis of real-world data have the potential to provide essential information on tumor samples before and upon treatment, which will help to decipher the molecular mechanisms responsible for resistance and guide the next-line therapies [3]. Research on cancer evolution has revealed new concepts on how cancer cells evolve due to drug treatment and actively develop resistance [4]. One proposed mechanism is that, in the initial macroevolution phase, dramatic changes induced by cancer drugs lead to genomic chaos and structural rearrangements, such as small-scale local alterations known as chromothripsis, multiple complex translocations called chromoplexy, chromosomal fragmentations, and the formation of polyploid giant cancer cells (PGCCs) [4,5,6,7]. In the second phase, stepwise microevolution mediated by gene mutations takes place [4,8]. Despite all the important discoveries made recently, this invaluable information, regrettably, often comes too late for many patients as the cancer has already spread to distant sites and become very heterogeneous due to the multiple and diverse processes involved in resistance. Anticipating acquired on-target and off-target resistance mechanisms already in preclinical models may allow to predict tumor liabilities early and therefore has the potential to guide the choice for single-agent or combination treatments likely to prolong therapy response (Figure 1b).

Multiple strategies have been used to predict and investigate the molecular mechanisms underlying drug resistance, mainly based on directed evolution approaches [9] (see Table 1). Early procedures were based on random mutagenesis of the drug target to identify modifications responsible for treatment resistance [10,11]. Another strategy is to treat tumor cells grown in vitro with increasing drug concentrations and thereafter analyze the surviving cell clones for molecular changes [12]. Preclinical in vivo studies, especially with patient-derived xenograft (PDX) models, have also successfully been performed to identify drug-resistant daughter cells and explore the mechanisms involved [13]. More recently, systematic strategies have been developed for an unbiased approach. One example is detailed mutational scanning of the drug target to determine which positions are essential for oncogenic activity or associated with treatment resistance [14]. A major limitation of this procedure is that it only addresses on-target and not off-target resistance mechanisms. Broader strategies aiming at a genome-wide investigation have gained increased interest. Engineered zinc finger nucleases (ZFN) and transcription-activator-like effector nucleases (TALEN) that allow to introduce DNA changes based on protein/DNA recognition paved the way for large-scale efforts towards genome editing [15]. However, these approaches necessitate the design of nucleases that specifically recognize the selected DNA sequence and have only modest efficiency, which limits their application. The field was much expanded by advances based on the bacterial clustered regularly interspaced short palindromic repeats (CRISPR) editing system, leading to simple and highly effective genome-editing technologies [16,17]. It uses individual guide RNAs to direct the nuclease with high accuracy toward the genomic region to be modified. CRISPR/CRISPR-associated (Cas) technology is now widely adopted for controlled gene editing in multiple model organisms and in human cells. Different variations of CRISPR/Cas enable an unbiased global screening of off-targets responsible for therapy resistance due to changes in gene expression and protein sequence, or in epigenetic signatures, so that pathways likely to represent novel treatment strategies can be identified [17,18,19].

We performed an extensive literature search to review the strategies currently used to determine on- and off-target resistance mechanisms to anti-cancer treatments. Keywords including cancer drug resistance, mutational scanning, CRISPR, and base editing were used to interrogate the MEDLINE database, with a focus on articles dealing with solid tumors and published between 2018 and 2023.

## 2. Non-Systematic Preclinical Anticipation of Drug Resistance Mechanisms

### 2.1. Random Mutagenesis

The era of targeted cancer therapy was heralded by drugs binding to the estrogen receptor (ER) or androgen receptor (AR), and later by the emergence of small-molecule kinase inhibitors [20,21,22]. Resistance mutations in the addressed target were, however, soon identified in treated patients. Early examples include point mutations in the ER and the AR in treated breast and prostate cancer patients, respectively, in the epidermal growth factor receptor (EGFR) in non-small-cell lung cancer (NSCLC) patients, and in kinases of the rapidly accelerated fibroblast (RAF)/mitogen-activated protein kinase (MAPK)/extracellular signal-regulated kinase (ERK) pathway in melanoma patients [23,24,25,26,27,28,29,30,31].

Random mutagenesis using different strategies was instrumental in predicting treatment resistance (Table 1). Examples include mutant libraries devised by error-prone polymerase chain reaction (PCR), mutated human immunodeficiency virus-1 (HIV-1) reverse transcriptase, or the use of bacterial strains with impaired DNA repair machinery, which have been successfully used to determine resistance mechanisms taking place following targeted therapy [32,33,34]. Mutations leading to the loss of inhibitor binding or even to the switch to activator properties have thereby been found. This has, for instance, been reported for the AR in prostate cancer [35,36]. A randomly mutated plasmid library expressing the AR co-transfected with a reporter gene assay was successfully used to identify amino acids leading to resistance to the AR inhibitor enzalutamide following selection by fluorescence-activated cell sorting (FACS) [37]. A random mutagenesis screen allowed the identification of amino acid residues in EGFR that lead to resistance to the CL-387,785 inhibitor in lung cancer cells H1975 [32]. Mutations conferring resistance to RAF kinase inhibitors PLX-4720 and RAF265, and affecting dimerization were identified in two different regions of cellular RAF (c-RAF) [10]. Various point mutations leading to reduced response of A375, WM266.4, and SKMEL-19 melanoma cells to MAPK inhibitors VX-11e, trametinib, and dabrafenib were found in ERK1 and ERK2 [11]. A random mutagenesis screen performed on MAPK/ERK kinase (MEK) in A375 melanoma cells grown in the presence of allosteric inhibitor AZD6244 revealed most resistance mutations to occur in the allosteric drug-binding pocket or in the alpha-helix C [34]. EGFR, BCR-ABL, and KRAS mutant libraries generated by techniques relying on HIV-1 reverse transcriptase errors allowed to pinpoint resistance mutations to the corresponding inhibitors gefitinib, imatinib, and sotorasib, respectively [33].

Random mutagenesis represents a comparably straightforward and robust method that does not require the design of large dedicated libraries and has the advantage of identifying base insertions and deletions [38,39,40]. It has proven beneficial in determining target mutations specific for the inhibitor applied during the experiment [10,11,32,33,34,37]. These mutations often also affect the efficacy of other compounds addressing the same target so that the findings may help for the synthesis of next-generation compounds. Disadvantages include the facts that coverage may be challenging for large proteins and that random mutagenesis libraries may have biases so that not all possible mutations are necessarily represented [41,42,43,44]. Also, mutation co-occurrence is not excluded within individual constructs and cannot be controlled, which complicates the identification of the decisive resistance alteration. Importantly, selecting the best assay(s) for the quantitative determination of the function of interest of the generated mutants will be crucial for a high-quality output [45].

### 2.2. Chronic Exposure of Tumor Models to Drug Regimens

A well-established strategy for uncovering the mechanisms responsible for drug resistance is the long-term treatment of tumor cells with an inhibitor and subsequent selection of surviving clones, followed by a detailed molecular characterization (Table 1). Patient-derived cell lines were originally examined due to their general availability, ease of use, and amenability to high-throughput analysis [46,47,48,49,50,51,52]. However, they only represent one cell type, which provides inherent limitations to the conclusions that can be drawn. Successful examples include in vitro experiments performed with EGFR tyrosine kinase (TK) inhibitors, which outlined the role of target mutation and of MET amplification in NSCLC PC-9 cells [46], as well as of Src kinase activation in esophageal squamous cell carcinoma cells [47]. Different cellular models of melanoma becoming resistant to BRAF inhibitors were found to harbor BRAF mutations and numerous growth factor pathway alterations [48]. In one study, primary melanoma cells (M160915) and established cell lines (A375, WM983A, UACC903) with resistance to the BRAF inhibitor dabrafenib were analyzed and it was found that the AR and its downstream pathway were upregulated [49]. Several groups have explored the acquired resistance to cyclin-dependent kinase (CDK) 4/6 inhibitors. In the EFM119 breast cancer cells, the resistance to palbociclib is linked to activation of the phosphatidylinositol 3-kinase (PI3K)/AKT/mammalian target of rapamycin (mTOR) pathway [50], whereas, in ovarian cancer cell lines such as ID8 and HOC7, the impact of the MAPK pathway was additionally evidenced [51]. Hormone-resistant breast cancer MCF-7 cells showed impaired double-stand break (DSB) repair when acquiring resistance to the poly ADP-ribose polymerase PARP inhibitor olaparib [52].

Studies performed with compounds that hijack the cellular proteasome system to degrade the bound protein, the so-called proteolysis-targeting chimeras (PROTACs), have shown promising results with regard to overcoming acquired resistance to classical inhibitors that stoichiometrically bind to their target and only block its function [53,54]. However, dedicated screening campaigns have now revealed that tumor cells can escape treatment with PROTACs via mutations in the recruited E2 and E3 proteasome complexes. For example, chronic treatment of acute myeloid leukemia (AML) MV4-11 cells with increasing concentrations of bromodomain and extra-terminal (BET) PROTACs recruiting the Von Hippel–Lindau (VHL) or cereblon (CRBN) E3 ligase led to the selection of resistant cells where the targeted protein had regained its original levels due to impairing mutations in the recruited ubiquitin ligase complexes [55]. Similarly, mutations in the core components of the VHL or CRBN E3 complexes were observed following prolonged exposure of AML (SKM1, MV4-11), ovarian cancer (OVCAR8), or prostate cancer (LNCaP) cell lines to BET PROTACs [56]. Another study where the ovarian cancer cell line A1847 underwent a long-term exposure to different BET or CDK9 PROTACs showed the role of the drug efflux pump MDR1 in both intrinsic and acquired resistance [57]. In vitro approaches with organoids and three-dimensional co-cultures potentially better mimic the real-life patient situation but are technically more challenging and generally permit less throughput [58,59].

In vivo experiments to evaluate cell lines or PDX models may better reflect the clinical cancer landscape for the determination of resistance mechanisms [60,61]. For instance, growing the prostate cancer LNCaP cells in castrated mice and treating them with the AR inhibitor enzalutamide shows that elevated glucocorticoid receptor (GR) expression is involved in the resistance process [62]. Another example is the in vivo analysis of different prostate cancer PDX models for resistance to the AR signaling inhibitors abiraterone and enzalutamide, which highlights expression of the constitutively active AR-V7 splice variant as a recurrent event [63]. Melanoma PDX models grown in mice in presence of the MAPK inhibitors trametinib and vemurafenib reveals the implication of DNA repair defects in resistance [64]. In vivo analysis of a melanoma PDX model with acquired resistance to the BRAF inhibitor vemurafenib shows strong expression changes in a few genes but no activation of canonical pathways [65]. Resistance of the NSCLC PDX model LXFA 677 to the EGFR inhibitor gefitinib involves mutations and stimulation of several genes, as well as activation of the STAT1, STAT3, MEK1/2, and NF-κB pathways [66]. A chemotherapy regimen mirroring the established neuroblastoma treatment was administered to mice harboring corresponding PDX models with MYCN amplification and this led to the identification of decreased adrenergic phenotype and enhanced immature mesenchymal-like cellular state as acquired resistance mechanisms [67]. The high-grade serous cancer models PDX-2428 and PDX-2462 transplanted onto mice and that develop resistance to PARP inhibitor olaparib exhibit a high enrichment in PGCCs with features of senescent cells [68].

A general drawback of these approaches is that tumor heterogeneity and evolution over time cannot be reflected properly in the models used [69,70]. Also, the impact of the complex tumor environment and competition for space are not considered [71,72]. Combining the experimental results with patient data will therefore be essential to maximize the value of the findings.

## 3. Systematic Preclinical Anticipation of On-Target Drug Resistance Mechanisms

### 3.1. Deep Mutational Scanning (DMS)

One strategy to enable the systematic discovery of on-target resistance mutations is DMS, which, unlike the methods described above, is a very comprehensive approach based on specifically designed libraries (Table 1). All possible single amino acid substitutions at all positions of the protein of interest are covered and the corresponding lentivirus expression library is then transduced into appropriate cells [14,45,73,74]. Functional assays to determine target protein binding to other biomolecules, stability, or activity can then be performed. Alternatively, phenotypic assays in presence of the selected inhibitor to enrich for resistance variants under conditions best reflecting the physiological situation are frequently used. Importantly, these assays can also be implemented under high-throughput conditions. Surviving cell clones are harvested and deep-sequenced before implementing a thorough data analysis to identify the individual on-target resistant mutations that led to drug escape (Figure 2). High-throughput sequencing and quantification of the phenotypes will allow measurement of the functional consequences of up to hundreds of thousands of protein variants simultaneously. Examples include mutagenesis studies performed to detect changes leading to resistance to different kinase inhibitors. A DMS analysis performed in the melanoma Meljuso cell line with the ATP-competitive CDK4/6 inhibitor palbociclib uncovered hotspot residues in CDK4 and CDK6 involved in resistance, which were also confirmed in ERK2 [75]. DMS was also conducted to comprehensively catalogue KRAS missense mutations conferring resistance to two KRAS^G12C^ inhibitors, MRTX1257, an adagrasib analogue, and sotorasib. Not surprisingly, most of the identified resistance mutations were at positions associated with drug binding, but additional mutations were located outside the drug-binding pocket of MRTX1257 and sotorasib [76]. Another study explored the sensitivity of EGFR variants to different approved inhibitors by implementing a cytotoxicity screen in the NSCLC cell line PC-9 depleted of endogenous EGFR and transduced with a lentiviral mutant library, and numerous common but also drug-selective variants were reported [77].

Generating a DMS library has high costs, and the screening campaign is labor-intensive. Limitations include the size of the library, which needs to include as many mutations as possible, and the transformation efficiency of the cells to cover the library size [74,78]. While it is usually feasible to include all single amino acid mutations of a protein of interest within one or two libraries, comprehensive covering of double- or multiple-amino-acid mutations of a target is not feasible [74]. Cell proliferation and cytotoxicity are often used as phenotypic endpoints [74]. Alternatively, pooled screens based on FACS and subsequent isolation of subpopulations providing the appropriate signal have also been tested [78,79]. Both approaches will, however, only reflect the complex environment found in tumors imperfectly. Also, spontaneous mutations in compensatory pathways involved in cell growth and survival cannot be excluded [9]. However, the DMS approach has the potential to identify numerous single-amino-acid on-target resistance mutations, including those located distantly from the inhibitor-binding pocket, so that novel important functional regions in the targeted protein may be uncovered. DMS can also reveal mutations that are rare in patients, thus enlarging the screening options in patient samples and improving the clinical interpretation of disease-associated drug target mutations. On the other hand, it will not inform about the frequency of the mutations in the clinic, which are known not to occur with equal probabilities in patients. Single-nucleotide variants (SNVs) are much more likely to arise than multi-nucleotide variants (MNVs) [80,81,82] and are not all equally likely to be represented. Prioritizing changes based on the mutational processes undergone by the patient cohort of interest can help to pinpoint which alterations are most likely to occur in the clinic, and consequently guide the clinical development strategies for next-line combination therapies. It is also worth noting that generating an atlas of on-target resistance mutations for a given drug will help to identify the population most likely to respond and avoid treatment of patients already harboring resistant mutations.

### 3.2. CRISPR-Based Base-Editing Screens

The CRISPR/Cas9 system has revolutionized biomedical research by allowing specific, efficient, and relatively easy manipulation of genomes of living cells and organisms. The concept originates from prokaryotic adaptive immunity mechanisms that protect bacteria from phage infections by cleaving the nucleic acids of the invading pathogens upon reinfection [83]. Nowadays, thanks to the discovery that a similar DNA cleavage can be programmed [84] and reconstituted in eukaryotic cells [85,86,87], CRISPR/Cas9 and its adaptations serve as precise, robust, and versatile tools for genome editing. A variety of delivery strategies including microinjection, electroporation, nanoparticles, extracellular vesicles, or viral-based methods have been evaluated to improve efficacy of gene-editing tools, as reviewed elsewhere [88,89,90].

CRISPR base editing is a genome-editing method that allows creation of a point mutation at a precise genomic locus without the need to form DSB nor the requirement of a donor template [91]. CRISPR-based base editors (BEs) take advantage of a dead Cas9 (dCas9) variant that does not cleave DNA, or, more often, a nicking Cas9 (nCas9) that produces single-strand DNA breaks. The technology also necessitates a deaminating enzyme that modifies a DNA base and a single-guide RNA (sgRNA) that navigates the modified Cas9 protein to a specific locus [92]. By now, two main types of BEs have been established: cytosine base editors (CBEs), mediating a C to T conversion on the targeted strand and consequently a complementary G to A conversion on the reverse strand [93], and adenosine base editors (ABEs) converting A to I, which is then replaced with G on one strand, followed by a T to C change on the other strand [94]. Since the first introduction of the technology, multiple BE variants were developed with much enhanced editing efficiency and specificity, as well as reduced off-target activity [93,95,96,97,98,99,100,101]. Moreover, dual-deaminase CRISPR BEs concurrently introducing both A-to-G and C-to-T mutations on the targeted strand were also designed and proven effective to substitute bases, as expected [102,103,104]. CBEs and ABEs generate transition mutations, which means changes from purine-to-purine or pyrimidine-to-pyrimidine. Substantial efforts have also been undertaken to enable single-base genomic transversions, which include purine-to-pyrimidine or pyrimidine-to-purine mutations. These efforts were successful with the help of glycosylase base editors (GBEs), which can induce a C to G and the corresponding G to C transversions [105,106,107]. The BE techniques are expanding dramatically and may ultimately be utilized for therapeutic strategies where pathogenic gene variants are corrected in a precise and safe manner [108].

In the context of anti-cancer drug resistance investigation, the BE technology can be applied to identify mutations that alter drug sensitivity (Table 1). The screen setup is like that of CRISPR screens described below in Section 4.1 and Section 4.2 but with a different library being used in which sgRNAs tile over one or several genes to be mutated. For example, BE3, a third-generation cytosine BE, was successfully used in the myeloid leukemia HAP-1 cells to generate and then functionally analyze *BRCA1* variants and uncover gene positions where substitutions confer resistance to the PARP inhibitor olaparib [109]. Similarly, *BRCA1*, *BRCA2*, and *PARP1* tiling sgRNA libraries were used for specific base editing to unravel mutations leading to resistance to a group of PARP inhibitors or to the chemotherapeutic drug cisplatin in various tumor cell lines using next-generation BE3 or BE4 CBEs [110]. The same study investigated synthetic lethal relationship of *MCL1* and *BCL2L1*, and uncovered mutations within these genes that confer resistance to their respective inhibitors S63845 and A-1331852 [110]. Another example of the usage of BE screening to anticipate resistance mutations is the tiling and mutagenizing of *BCL2* and *BRCA1* genes to find mutations leading to resistance to the B-cell lymphoma 2 protein binder venetoclax in A375 melanoma [111]. A recently published report first took a CRISPR knockout (CRISPRko) screening approach (see Section 4.1) in the colorectal cancer (CRC) cell lines HT-29 and LS-411N to identify mediators of sensitivity and resistance to interferon (IFN)-γ, a determinant of the response to checkpoint inhibitors, and then further used ABEs and CBEs to conduct mutagenesis of the top-hit genes. This allowed the discovery of essential mediators of sensitivity or resistance to IFN-γ, and, importantly, a strong overlap with available patient data was observed [112]. In another study, CRISPR-mediated screens with CBEs and ABEs targeting altogether nearly 18,000 phosphorylation sites were performed in the CRC HCT116 cells to understand resistance mechanisms to the anti-metabolite 5-fluorouracil, thus leading to the identification of RSK2 and PAK4 kinase substrates [113].

BE is a powerful and, compared to DMS, relatively cheap technique that can pinpoint positions conferring drug resistance in the gene(s) of interest. However, tiling libraries do not ensure representation of all possible substitutions at all gene positions. Another limitation is the inability to control and assess the off-target effects genome-wide as the methods for BE off-target detection are not sufficiently advanced yet.

### 3.3. Computational Methods

The applications of computational methods for predicting biological events have dramatically expanded by virtue of increasing computing power, general accessibility of a diversity of dedicated software, and the growing amount of available biological information, such as genomic, transcriptomic, and structural protein data. Computational methods are therefore also used for studying and predicting cancer drug resistance (Table 1). They can be divided into mechanistic modelling approaches and data-driven computational prediction methods [114]. Molecular dynamics (MD) simulations are an example of the first category. Their aim is to determine how individual atoms and molecules move over time and, in the context of resistance, predict how mutations perturb protein structure or impact drug-binding affinity [115,116]. MD simulations were, for instance, applied to provide molecular insights into resistance to the TK inhibitor imatinib, a drug used to treat different leukemia forms and gastrointestinal stromal tumors [117]. This method was also chosen to predict sensitivity or resistance of EGFR exon 20 insertion mutants to osimertinib, a third-generation TK inhibitor approved to treat NSCLC [118]. Machine learning has furthermore been applied to predict resistance mechanisms. It is a branch of artificial intelligence (AI) and exploits input data to train a model that is then used to make predictions. Machine learning algorithms have, for example, allowed to anticipate the resistance of breast cancer patients to the chemotherapeutic drug doxorubicin based on miRNA profiles [119] and to predict response to neoadjuvant therapy in early breast cancer [120]. They have also successfully been applied to predict response to anti-PD-1 immunotherapy in melanoma and NSCLC patients [121]. In another investigation, 1001 cancer cell lines were used to train networks and forecast drug response, followed by a comparison with data from clinical cohorts, which confirmed the predictive value [122]. Altogether, the recent advancements in machine learning are much helped by the growing amount of pharmacogenomic data derived from patient-relevant tumor models and will ultimately lead to improved predictions [123]. Bringing together small-scale mechanistic modelling approaches with large-scale machine learning predictions has the potential to leverage our current knowledge and more precisely predict individual patient response to a given treatment [124]. In 2022, MD simulations together with machine learning algorithms were exploited to build a personalized drug response prediction model to prognosticate the response of lung cancer patients to the EGFR TK inhibitors gefitinib and erlotinib [125]. AI models, trained with systematic and good-quality data, may one day become sufficiently accurate to substitute sophisticated, expensive, and time-consuming wet-lab experiments. However, for now, findings predicted by computational tools, including the anticipation of resistance mutations, still require biological validation.

## 4. Systematic Preclinical Anticipation of Off-Target Drug Resistance Mechanisms

### 4.1. CRISPR/Cas9 Knockout Screens to Identify Genes and Pathways That Confer Drug Resistance

As mentioned above, CRISPR/Cas9 strategies are now frequently adopted for conducting comprehensive genetic screens in order to investigate gene function [17,59,85,126,127,128,129,130,131,132]. In the widely used knockout procedure, the Cas9 nuclease and a sgRNA precisely directing Cas9 to the targeted location are delivered into the cell of interest. Binding of the sgRNA necessitates a protospacer adjacent motif upstream of the targeted site, which is frequently found in the human genome [90]. Once delivered, Cas9 creates a DSB at a location specified by the co-administered sgRNA, which, in the absence of homology-directed repair (HDR) template, is usually repaired by error-prone non-homologous end-joining (NHEJ) [85,133]. This often results in a short DNA insertion or deletion (indels), leading to frameshift mutation and premature stop codon introduction and consequently permanent inactivation of the targeted gene [134]. Gene disruption using the CRISPR knockout (CRISPRko) system much accelerated the understanding of gene function in normal conditions as well as in the disease state [87,132] (Figure 3a). Due to its simplicity and efficacy, this technology allows to perform multiple gene knockouts simultaneously in one cell or even in a large cell population in a single experiment [132]. From 2012 onwards, CRISPRko has led to major advances in the identification of cancer-driving events, both in coding and non-coding genome regions [59].

Conducting a genome-wide CRISPRko screen while exposing the cells to an anti-cancer drug has the potential to systematically uncover the genes associated with resistance to the given treatment (Table 1). The experimental design usually starts with the selection of an appropriate cell model and generation of the corresponding Cas9-expressing stable cell line. These engineered cells are then transduced with a sgRNA library and challenged with the compound of interest to select for resistant cells. A sequencing-based readout will subsequently allow to compare sgRNA frequencies in the control and treated conditions, and to find the genes conferring resistance (Figure 3b). Exploring mechanisms of resistance to cancer medications using genome-wide lentiviral sgRNA libraries was first proposed by two independent groups in late 2013 [135,136]. The first study confirmed the feasibility and accuracy of such a screening strategy by transducing HL-60 and KBM-7 leukemia cells and subsequently challenging them with the purine antimetabolite 6-thioguanine or the topoisomerase II inhibitor etoposide [135]. Not surprisingly, the CRISPR/Cas9-based resistance screen revealed that members of the DNA mismatch repair pathway are involved in resistance to 6-thioguanine. It furthermore showed that loss of *TOP2A* confers strong protection against etoposide and also identified the role of *CDK6* in mediating drug cytotoxicity [135]. The second study searched for the cause of resistance of A375 melanoma cells to the BRAF inhibitor vemurafenib [136]. Genes previously reported to be involved were highlighted, namely *NF1* and *MED12*. In addition, novel hits such as *NF2*, *CUL3*, *TADA2B*, and *TADA1* were identified [136]. A saturation mutation CRISPRko study focusing on *cis*-elements identified regulatory regions in different genes involved in vemurafenib resistance in A375 melanoma cells [137]. Since then, many additional studies have been performed in tumor cells using the CRISPRko technology to detect resistance mechanisms, mainly for cytotoxic drugs [126,131]. Concerning targeted therapies, a genome-wide CRISPRko screen carried out in breast cancer cells derived from MCF-7 developing resistance to different ER inhibitors uncovered the role of purine biosynthesis [138] and of the epigenetic modulator ARID1A [139]. MCF-7 breast cancer cells were also screened for resistance mechanisms to the PI3Kα inhibitors alpelisib and taselisib, and negative regulators of mTORC1 were thereby identified [140]. Resistance of the prostate cancer cells LNCaP to the AR inhibitor apalutamide is linked to *TLE3* loss, and this is also the case for resistance to enzalutamide, a structurally close AR inhibitor, as shown in subsequent validation experiments [141]. Another CRISPRko screen performed with enzalutamide in the C4 prostate cancer cell line demonstrated that *IP6K2* knockout increased resistance, whereas *XPO4* knockout had the opposite effect [142]. A CRISPRko screen to understand enzalutamide resistance in prostate cancer CWR-R1 and 22Rv1 cells and focusing on kinases revealed the role of activated BRAF [143] and of CK1α [144]. Importantly, the role of CK1α was confirmed in the LuCaP35CR and LuCaP77CR PDX prostate cancer models. Resistance of several prostate cancer cell lines to the PARP inhibitor olaparib is linked to CHEK2 loss [145]. Another study performed in C4 cells documented that resistance to olaparib correlates with loss of genes such as *PARP1*, *ARH3*, *YWHAE*, or *UBR5* [146]. Multiple CRISPRko screens were implemented to identify resistance mechanisms to PARP inhibitors in additional tumor cell lines, thus allowing to better understand the regulation of different DNA damage response (DDR) pathways and their crosstalk [147]. Cell cycle control and replication stress were found to be essential players [147]. Another genome-wide screen was carried out in the NSCLC HCC287 cells to determine the mechanisms responsible for resistance to the EGFR TK inhibitor erlotinib. The YAP signaling pathway and the cullin-5 E3 complex were associated, negatively and positively, respectively, with resistance [148]. In another study performed in NCI-H820 lung cancer cells developing resistance to erlotinib, gene signatures from the cell division cycle and protein ubiquitination process were identified, and inhibition of either pathway led to comparably increased anti-tumor efficacy in vivo [149]. Resistance gain of KRAS-mutated HCT116 colon cancer cells to the MEK inhibitor selumetinib involves the *GRB7* gene, which encodes a protein interacting with receptor TKs [150]. A kinome-centered CRISPRko screen performed in the hepatocellular cancer (HCC) SNU449 cells highlighted the implication of EGFR in resistance to the multi-kinase inhibitor lenvatinib [151]. Another screen performed in the lung squamous H520 cells and focusing on the kinome showed that polo-like kinase (PLK) 1 mediated resistance to fibroblast growth factor receptor (FGFR) inhibitors such as AZD4547 and BGJ398 [152]. A CRISPRko screen conducted in the rhabdoid tumor cells G401 and G402 in presence of the enhancer of zeste homolog 2 (EZH2) inhibitor GSK126 revealed that the H3K36me2 methyltransferase nuclear receptor binding SET domain protein (NSD) 1 was connected with resistance [153]. This was mechanistically linked to the role of NSD1 in cooperating with the SWI/SNF and PRC1/2 complexes for transcription activation and differentiation. In 2023, a study performed in multiple tumor cell lines aiming to anticipate resistance mechanisms to an inhibitor of SHP2, an important activator of RAS, reported the role of expected resistance genes but also of novel players such as *INPPL1*, *MAP4K5*, and *LZTR1*, which was confirmed in vivo [154]. Concerning resistance to PROTACs, CRISPRko screens completed in MM.1S myeloma cells allowed to determine which gene losses mediate resistance to CRBN- and VHL-based bifunctional degraders. The top hits were genes coding for proteins from the degradation machinery itself and not the ones associated with the targeted oncoprotein [155]. Another recent CRISPRko screen performed in H1299-7 NSCLC cells investigated the mechanisms responsible for resistance to targeted radionuclide therapy and established that genes involved in binding and retention of the drug tested and in the DDR pathway were essential players [156]. Resistance mechanisms to immune checkpoint inhibitors have also been determined by CRISPRko. These drugs have significant efficacy in many tumor types, but, here, resistance mechanisms also ultimately develop. Several dedicated CRISPRko screens have been performed to better understand the mechanisms involved, and they have recently been reviewed [157,158,159].

A few CRISPR screens have also been performed in vivo with xenograft models. The circulating tumor-cell-derived breast cancer metastasis model CDX-BR16 was used in a screen based on a loss-of-function CRISPR approach and *IL18R1*, *ITGA2*, *CSNK1A1L*, and *CSNK2A2* were identified as playing a role in the metastatic burden, and *PLK1* as involved in intravasation [160]. A genome-wide CRISPRko in vivo screen performed in the triple-negative breast cancer (TNBC) model SUM159PT highlighted the important role of the mTOR and the YAP pathways in tumorigenesis [161].

In summary, multiple reports have now shown that CRISPRko screens allow to identify genetic determinants of resistance to approved or advanced anti-cancer therapeutics in an unbiased manner. Such screens have been applied to model the progressive acquisition of oncogenic events and determine the impact of multiple mutations [59]. However, this approach has limitations, such as the occurrence of off-target effects that may take place at one out of 1000 to 13,000 targeted sites, depending on the sgRNA used [162,163,164]. Also, formation of micronuclei and chromosome bridges potentially leading to chromotrypsis and large genomic rearrangements have been reported in many cell lines edited by CRISPR/Cas9, which ultimately can induce off-target editing [165]. A high number of micronuclei and chromosomal aberrations was, for instance, reported in a CRISPRko study targeting thirty-four tumor suppressor genes for induction of liver tumors in mice [166]. New developments to reduce such off-targets include utilization of improved Cas homologs, adjustments in sgRNA length and chemical changes, use of base editors, as well as advancements in the Cas9 and sgRNA administration route [90,167,168,169]. Unspecific DSB may occur especially at amplified loci and lead to false positives due to cytotoxicity and apoptosis [170,171,172]. Other shortcomings are variations in the mutational outcomes [173], the reliance on complete knockout as opposed to partial inactivation to generate a hypomorphic phenotype [174], and the fact that non-coding RNAs and highly repetitive sequences are not easily addressed [175,176].

### 4.2. CRISPR Interference (CRISPRi) and CRISPR Activation (CRISPRa) Screens to Reveal Gene Expression Modulations That Confer Drug Resistance

Over the past decade, the CRISPR technology has progressed rapidly and now uses a variety of modified Cas nucleases with distinct requirements, off-target profiles, and cleavage specifications for improved genome editing [17,128,157]. A significant discovery was that dCas9 lacking endonuclease activity due to inactivating mutations in its two nuclease domains is still capable of binding to sgRNA and the targeted DNA strand region [177,178]. Provided that targeting occurs in the transcribed region, the dCas9-sgRNA complex will block RNA polymerase activity and transcription elongation, consequently leading to highly specific gene silencing. Alternatively, targeting regulatory elements such as promoters or enhancers will also lead to effective gene silencing [59,178]. This efficient and selective strategy to repress target gene transcription with the help of dCas9 is named CRISPRi (Figure 3a). An additional advancement is the fusion of dCas9 with a transcriptional inhibitor, such as the Kox1 domain of the Krüppel-associated box (KRAB), known to recruit chromatin repressor complexes, which strongly inhibits gene expression, in some instances by 90% or more [16,179]. The same study also shows that linking dCas9 with a transcriptional activator such as VP64 leads to stimulation of gene expression, thus paving the way for CRISPRa approaches [16,179] (Figure 3a). This is achieved via activation of silent genes or upregulation of already active genes, but the stimulation is usually not very pronounced. Following these initial findings, the CRISPRi/a technology has undergone significant improvements, especially regarding the magnitude of effects achieved. New repressor domains such as KOX1-KRAB-MeCP2 or ZIM3-KRAB [180,181] and second-generation programmable transcriptional activators such as the SunTag array, the tripartite effector VP64-p65-Rta (VPR), and the synergistic activation mediator (SAM) [182,183,184] are substantially much more effective tools for gene expression silencing or upregulation, respectively. Epigenome modifiers affecting DNA methylation or histone modifications have also been fused to dCas9 to achieve selective and strong expression regulation following targeting of distal gene regulatory regions [127]. Repurposing the CRISPR/Cas9 genome-editing tool as a gene expression regulatory platform has undoubtedly opened novel avenues for genome-wide screens aiming at evaluating the impact of gene expression modulation on healthy and diseased cells, and also for the determination of the effects of drug treatment. The screening setup to uncover the regulation of which genes will confer resistance to a given anti-cancer treatment is analogous to the CRISPRko screen, the only differences being the Cas9 fusion variant selected and the genome-wide lentiviral library type used (Figure 3b).

CRISPRi loss of function (LoF) and CRISPRa gain of function (GoF) screens have successfully been implemented to discover drug resistance mechanisms to different anti-cancer drugs (Table 1). A triple scheme with a parallel CRISPRko, CRISPRi, and CRISPRa screen was performed in A375 melanoma cells to single-out genes responsible for resistance to vemurafenib [185]. Importantly, there was a strong overlap between the hits found in the CRISPRko and the CRISPRi screens, but higher sensitivity was reported for the CRISPRi approach, which is better suited to identify hypomorphic effects. The potential importance of *MED12*, *FOXD3*, and *MED23* was, for instance, reported. It is also worth mentioning that the knockout of *FOXD3* was found to confer resistance to vemurafenib in another CRISPRko screen performed in the A375 melanoma cells [186]. In many instances, there were complementary findings from the CRISPRi and CRISPRa screens; for example, *EGFR* and *IGTB5* were downregulated or upregulated, respectively, in the resistance screen. Comparing the results of three different screens performed in parallel clearly shows that some resistant phenotypes can only be observed under given experimental conditions. In line with this, some hits only found in the CRISPRi or the CRISPRa screen were confirmed by the fact that other hits from the same biological pathway were identified in the reciprocal screen. A dual LoF/GoF screen carried out in *NRAS*-mutated Meljuso melanoma cells treated with a combination of MEK1/2 and CDK4/6 inhibitors uncovered that activated KRAS was sufficient to confer resistance [187]. Another dual screen in EGFR-dependent PC9 lung cancer cells treated with erlotinib and the CDK7/12 inhibitor THZ1 showed that the suppression of multiple genes associated with transcriptional complexes enhanced the efficacy of the combination treatment [188]. A CRISPRi screen performed in the NSCLC LK2 cells revealed that the cullin-5 ubiquitin ligase complex mediates resistance to drugs targeting MCL1 directly (AZD5591) or indirectly (CDK9 inhibitor AZD5576) [189]. A CRISPR LoF screen completed in the MDA-MB-361 breast cancer cell line resistant to HER2 inhibition and focusing on genes involved in the main growth signaling and metabolic pathways identified the non-oxidative pentose phosphate pathway as being essential for survival [190]. A recent CRISPRi screen focusing on long non-coding RNAs (lncRNAs) differentially expressed in glioblastoma tumors revealed *DARS1-AS1* to play a key role in resistance to radiation therapy [191]. A genome-scale CRISPRa screen in metastatic castration-sensitive LNCaP prostate cancer cells reported the participation of the paired-related homeobox2 (*PRRX2*) gene in enzalutamide resistance [192]. This study also highlighted genes that had already been linked to enzalutamide resistance, like *AR*, *WNT1*, *ID1*, and *ID3*. Another genome-wide CRISPRa screen in the bladder cancer cell line T24 acquiring resistance to the CDK4/6 inhibitor palbociclib detected the implication of multiple growth factor signaling pathways [193]. Resistance of MCF-7 breast cancer cells to palbociclib is linked to stimulation of cyclin E/CDK2 and Myc activity, as evidenced by CRISPRa experiments [194]. Another CRISPRa screen performed in BCL-2-expressing *Eµ-Myc/dCas9a-SAM^Kl/+^/sgBcl-2* lymphoma cells showed the BCL-2-related A1 protein to be associated to resistance to venetoclax [195]. A CRISPRa screening approach was also applied to activate expression of lncRNAs in order to identify putative players with a role in vemurafenib resistance in A375 melanoma cells, and loci potentially involved have been uncovered [196]. Activation of one of the validated hits, the *EMICERI* locus, leads to stimulation of several neighboring genes, including *MOB3B*, which encodes a kinase activator from the Hippo signaling pathway. Another work assessing vemurafenib resistance of melanoma cells derived from SKMEL-239 also focused on activation of lncRNA loci and listed four candidates as being implicated [197].

In summary, the CRISPRi/a strategy has significant advantages over other technologies for the unbiased elucidation of drug resistance mechanisms. Firstly, in comparison to CRISPRko, CRISPRi better mimics therapy as drug treatment often only reduces gene expression rather than entirely shutting it down. It also allows studying essential genes, which is not possible with CRISPRko due to the lethality of a complete knockout. In comparison to RNAi (reviewed in [198,199]), CRISPRi has fewer off-target effects, and the gene expression downregulation lasts much longer, thus allowing larger assay windows for cell exposure to the drug of interest. CRISPRi is also well-suited to target non-coding RNAs, including miRNAs, which are often redundant and can be addressed effectively with a multiplexing procedure [200]. CRISPRa offers advantages over an alternative exogenous over-expression of the protein of interest as it allows genome-wide coverage and the study of upregulation of very long transcripts. However, it was reported that the level of activation by CRISPRa depends on basal expression level and chromatin status [201], which may lead to biased results. CRISPRa may also identify pathways not detectable by an LoF approach due to cellular redundancy. A parallel CRISPRi/a approach has the potential to find genes with opposite phenotypic impacts, thereby providing an internal confirmation likely to accelerate the selection process for truly positive hits [126]. It will also be informative in case a gene involved in drug resistance only reveals its role when being either over- or under-expressed, for instance when the encoded protein is part of a multi-subunit complex [17]. Variations in tumor heterogeneity can also be followed by determining enriched cell clones over time and performing retroactive lineage tracing [59]. A remaining problem to be solved is that off-target detection methods for CRISPRi/a are not well-established yet, while for CRISPRko there is a wide range of possibilities, such as GUIDE-seq, BLISS, or DISCOVER-Seq+ [202,203,204]. Possibly, new developments in artificial intelligence and deep learning will ameliorate predictions [205].

## 5. Discussion

Non-systematic approaches such as random mutagenesis of targeted proteins and selection of surviving cell clones were originally used to identify on- and off-target resistance mechanisms to cancer drugs (Table 1). Following that, the comprehensive and versatile CRISPR/Cas9 technology has opened up broader prospects for precise manipulation of DNA and dissection of cell function. There are now multiple applications of CRISPR screen in cancer, including the exploration of mechanisms responsible for tumor growth and metastasis, and those involved in drug response and resistance. The latest developments in CRISPR/Cas-derived genome editor approaches have been instrumental for the prediction of the molecular on- and off-target mechanisms underlying cancer drug resistance (Table 1). Dedicated portals collecting data from CRISPR screens focusing on resistance to specific treatments, for instance PARP inhibitors, are now available [147]. Technical advances using different Cas nucleases, more specific and efficacious guide RNAs, and improved delivery systems have been instrumental in enhancing the efficiency and specificity of this technology, and in making it less expensive [59,90]. Also, increasing the amount of cell lines or other model types analyzed and performing independent screenings has much improved output quality. Complementary LoF and GoF approaches in cell- and organoid-based models represent very promising approaches and will continuously ameliorate our knowledge of the complex processes involved and uncover novel actionable mechanisms. Especially, pairing with functional genomics data such as single-cell sequencing and spatial transcriptomics provides hope for major advances [206,207,208,209,210]. There are now a growing number of examples where a CRISPR/Cas-based screen allowed to identify pathways responsible for de novo or acquired drug resistance pathways, including activation of alternative pathways, chromatin reprogramming, and cell lineage switches [126,129,130]. Recently, CRISPR screenings have been performed in patient samples to guide subsequent treatments, mostly in hematological tumors where tumor samples are more easily accessible, but subsequent clinical translation has proven difficult [211].

Aside from allowing to anticipate drug resistance, the dramatic improvements achieved in the CRISPR system have also led to applications in personalized cancer therapy [19,59,212]. CAR-T cells have been successfully reprogrammed by precise genomic insertion to enhance anti-tumor efficacy and increase cell persistence, mainly regarding treatment of blood cancer. Numerous early clinical trials are currently ongoing to evaluate the safety and efficacy of this novel approach [213]. Another personalized procedure boosted by CRISPR is cancer gene therapy [214]. Examples include the inactivation of mutated oncogenes or mutation reversion of tumor suppressor genes. The technology has also been used to enhance the expression of tumor suppressor genes. Another recent progress is the CRISPR-based prime editing technology, which can generate insertions, deletions, or point mutations at specific locations and with high specificity to modify the phenotypes responsible for cancer development or therapy resistance. This approach has been beneficial for the generation of cancer organoid models with specific alterations and has the potential to reverse cancer-causing mutations [215,216].

In this review, we have compared non-systematic and systematic approaches aiming at identifying on- and off-target resistance mechanisms to cancer drugs. The main advances have recently been achieved in adaptations of the CRISPR technology, enabling comprehensive, large-scale, and complementary screens. In addition, coupling the CRISPR technology with cutting-edge bioinformatic tools has dramatically improved our knowledge on cancer initiation and progression, and also on therapy resistance, by elucidating the role of individual genes, enabling the identification of multiple interacting pathways, determining the part played by non-coding RNAs, and identifying numerous regulatory elements of oncogenes and tumor suppressor genes. The first examples of pathways and targets identified via these approaches and leading to clinical evaluation of novel drugs have now been reported, mainly in hematological tumors [211,217]. This provides hope for major advancements in the understanding and treatment of hematological and solid tumors in the near future.

## Figures and Tables

**Figure 1 ijms-25-00705-f001:**
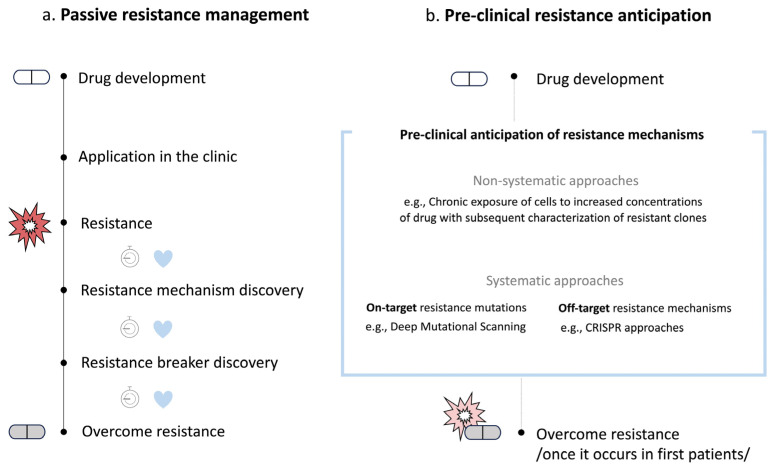
Types of resistance management. (**a**) In passive resistance management, efforts towards overcoming resistance start after loss of drug efficacy is observed in many patients. There is a long time between the occurrence of the resistance in the clinic and the identification of an appropriate resistance-breaking drug. (**b**) Preclinical resistance anticipation tries to predict on- and off-target biological events driving drug resistance before they occur in the first patients.

**Figure 2 ijms-25-00705-f002:**
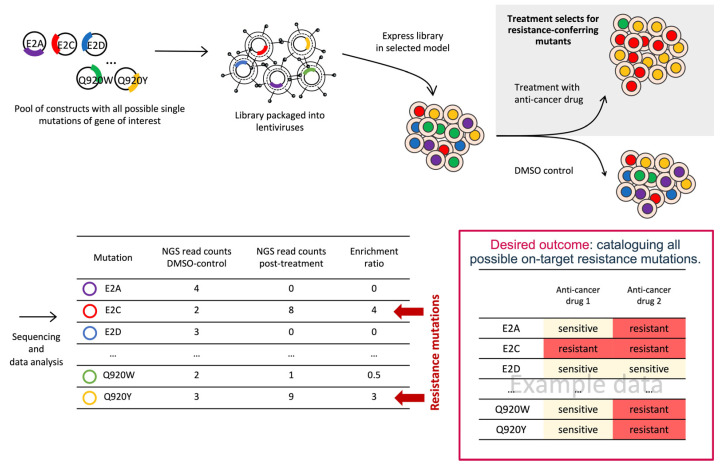
Deep mutational scanning for systematic discovery of on-target resistant mutations. The lentiviral library containing all possible single amino acid substitutions at all positions of the protein of interest is transduced and over-expressed in the selected cell model. Drug treatment selects for resistance-conferring mutants and the corresponding cells are harvested and deep-sequenced to identify the exact resistance mutation(s).

**Figure 3 ijms-25-00705-f003:**
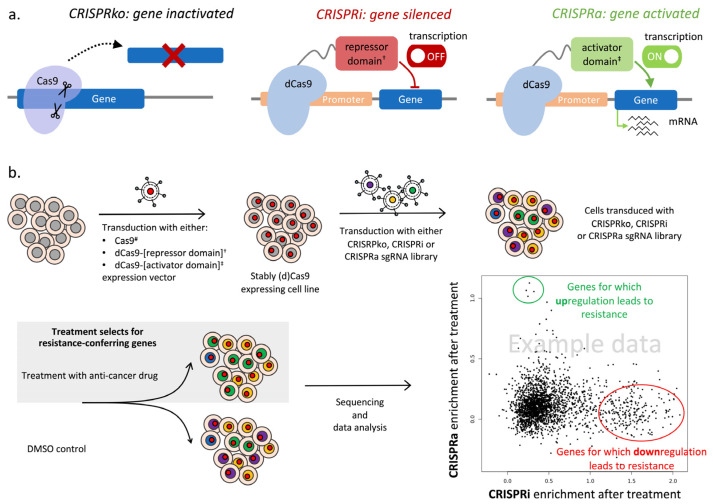
CRISPR screens for systematic discovery of off-target resistance mechanisms. (**a**) Cas9 variants used for different CRISPR approaches. In the CRISPRko approach, catalytically active Cas9 complexed with sgRNA cleaves the DNA within the targeted gene, consequently leading to its inactivation. In the CRISPRi and CRISPRa approaches, catalytically inactive dCas9 is fused with either a repressor domain or an activator domain and is guided to a specific gene regulatory region to either downregulate or upregulate the corresponding expression. (**b**) Preparation for the CRISPR-based screen usually starts with generation of a dCas9-expressing stable cell line. These engineered cells are then transduced with a genome-wide sgRNA library and challenged with the compound of interest to select for resistant cells. A sequencing-based readout allows to compare sgRNA frequencies in the control and treatment conditions, and to identify the genes conferring resistance. ^#^—or alternative, catalytically active Cas variant; ^†^ e.g., KRAB domain; ^‡^ e.g., SunTag, VPR, SAM.

**Table 1 ijms-25-00705-t001:** Different approaches to preclinically anticipate resistance mechanisms.

Type of Resistance	Approach	Method
On-target and/or off-target	Non-systematic	Random mutagenesis
		Chronic exposure of cells to increased concentrations and characterization of resistant clones
On-target	Systematic	Deep Mutational Scanning (DMS)
		CRISPR base editing (BE)
		Computational methods
Off-target	Systematic	CRISPR knockout (CRISPRko)
		CRISPR interference (CRISPRi)
		CRISPR activation (CRISPRa)

## Data Availability

Only previously published data are discussed.

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
