# Peer review of "Preclinical Anticipation of On- and Off-Target Resistance Mechanisms to Anti-Cancer Drugs: A Systematic Review"

_ijms, 2024, doi:10.3390/ijms25020705_

Round 1
Reviewer 1 Report
Comments and Suggestions for Authors
Preclinical anticipation of on- and off-target resistance mechanisms to anti-cancer drugs
Opinion: Approved with considerations
CONSIDERATIONS
LAYOUT
1. Please list references between [] throughout the text
2. Table 1 was not mentioned or discussed in the text
TITLE
1. Although the type of article is specified above. Suggest adding a term in the title indicating that it is a review and the type (e.g.: Preclinical anticipation of on- and off-target resistance mechanisms to anti-cancer drugs: a systematic review).
ABSTRACT
1. Add a short conclusion on the most promising emerging model reviewed by the authors.
1. INTRODUÇÃO
2. Lines 26 – 28: Add numerical data that emphasizes why these cancers are the most prevalent. "and now all have declining mortality rates", replace "now" with a more precise time range (Example: between 2014 and 2024...) and add numerical data that emphasizes how much this decline has been.
3. Lines 28 – 30: Even if it's rare, mentioning what treatments are current is interesting. Are they just chemical drugs? Are there invasive processes?
4. Include in the last paragraph the type of review being carried out (Literature, Integrative, Systematic or similar), if applicable adapt to the title as suggested in 1. under Title.
. Add a methodology topic that addresses how the review was constructed: Method of analysis, time range, inclusion criteria, number of articles selected, database, keywords.
2. NON-SYSTEMATIC, PRECLINICAL ANTICIPATION OF DRUG RESISTANCE MECHANISMS
2.1. Random mutagenesis
1. Lines 74 – 75: I suggest including the most classic types, or those cited in the literature, of mutations that generate resistance in the target, as well as including the types of target (proteins, receptors or correlates) most susceptible to resistance and why. If it gets too long, consider creating a separate topic before “2. Non-systematic, preclinical anticipation of drug resistance mechanisms”.
2. Lines 92 – 95: “Another non-systematic technology is to use lentiviral transduction...” if it's another technique found in the literature, then why hasn't it been covered with the same prominence? “2.1. Random mutagenesis”? It would be interesting to create a subtopic within “2. Non-systematic, preclinical anticipation of drug resistance mechanisms” addressing it in more depth.
a. If it is not well supported in the literature, then it is worth mentioning in the text the lack of papers dealing with it.
3. Lines 96 – 102: Are the statements regarding cost-benefit, use, and disadvantages reported in the researched literature or is it a conclusion of the authors? Please add references that address and emphasize these statements.
2.2. Chronic exposure of tumor models to drug regimens
5. Lines 110 – 113: Add references that confirm the statements.
6. When citing a tissue-specific cancer (e.g. breast cancer), please also cite the lineage used in the study to which the cancer belongs (e.g. MCF-7, 4T1, MDA-MB-231). Please consider this suggestion for the entire manuscript.
7. Lines 161 – 163: Add a reference to support your statement. Please consider this suggestion for the entire manuscript.
3. SYSTEMATIC, PRECLINICAL ANTICIPATION OF ON-TARGET DRUG RESISTANCE MECHANISMS
3.1. Deep Mutational Scanning (DMS)
8. Ok
3.2. CRISPR-based base-editing screens
9. Ok
3.3. Computational methods
10. Ok
4. SYSTEMATIC, PRECLINICAL ANTICIPATION OF OFF-TARGET DRUG RESISTANCE MECHANISMS
4.1. CRISPR-Cas9 knockout screens to identify genes and pathways that confer drug resistance
11. Line 333: It is interesting to modify "in recent years" by a specific and updated time range (e.g.: from 2022, CRISPRko has led...). Consider this suggestion for the entire manuscript and for other terms that refer to current dates without a defined period.
12. Line 380: “and inhibition of either pathway led to increased anti-tumor efficacy...”, how many % effective? It would be interesting to add the quantitative values of the findings to enhance the results. Consider this suggestion for other excerpts extolling better anti-tumor results.
13. Lines 387 – 390: “of the enhancer of zeste homolog 2 (EZH2) inhibitor GSK126 revealed that the H3K36me2 methyltransferase NSD2 was connected with resistance (135).”, for this and the other passages throughout the manuscript, which expose possible mechanisms of resistance or not, it is interesting to delve deeper into the approach of the results. Did the studies cited compare tumors/grafts? Did they carry out other tests involving the mechanisms? If so, when possible, add the comparative values that the authors found with the control groups during the pre-clinical phase.
4.2. CRISPR interference (CRISPRi) and CRISPR activation (CRISPRa) screens to reveal gene 418 expression modulations that confer drug resistance
14. Linha 460 – 663: “There were also findings specific to the method used, indicating that some phenotypes can only be observed under given experimental conditions” cite an example of a phenotype and condition. Is the reason for this event discussed, clarified or suggested? If so, I think it would be interesting to add it too.
5. DISCUSSION
15. It would be interesting to emphasize which of the techniques discussed in the manuscript proved to be the most promising, as well as to provide more details on their future prospects for widespread use.
16. Is the most promising technique the one most covered in the literature? If yes, is there a prevalence of studies addressing its pre-clinical use for which type of cancer/xenograft? If not, what prevents its further exploration to determine on- and off-target resistance mechanisms? Is it just the reasons highlighted at the end of the paragraphs of the reviews preceding the discussion?
CONCLUSION
17. Even if the logical reasoning has been closed in the discussion, there needs to be a conclusion highlighting the nature of the manuscript, the main findings, the most promising techniques and, among them, the technique that showed the best pre-clinical performance for further exploration in future pre-clinical studies.
Author Response
Many thanks for extensively reviewing our paper. Here how we addressed the individual points:
Review 1
Opinion: Approved with considerations
Please list references between [] throughout the text: Style of the brackets was changed
Table 1 was not mentioned or discussed in the text: The table is now better referred to in the Introduction, Results and Discussion sections
TITLE
Although the type of article is specified above. Suggest adding a term in the title indicating that it is a review and the type (e.g.: Preclinical anticipation of on- and off-target resistance mechanisms to anti-cancer drugs: a systematic review). Done
ABSTRACT
Add a short conclusion on the most promising emerging model reviewed by the authors. Done
- INTRODUCTION
Lines 26 – 28: Add numerical data that emphasizes why these cancers are the most prevalent. "and now all have declining mortality rates", replace "now" with a more precise time range (Example: between 2014 and 2024...) and add numerical data that emphasizes how much this decline has been. These details have now been added
Lines 28 – 30: Even if it's rare, mentioning what treatments are current is interesting. Are they just chemical drugs? Are there invasive processes? Details have been added
Include in the last paragraph the type of review being carried out (Literature, Integrative, Systematic or similar), if applicable adapt to the title as suggested in 1. under Title. Done
Add a methodology topic that addresses how the review was constructed: Method of analysis, time range, inclusion criteria, number of articles selected, database, keywords. Done
- NON-SYSTEMATIC, PRECLINICAL ANTICIPATION OF DRUG RESISTANCE MECHANISMS
2.1. Random mutagenesis
Lines 74 – 75: I suggest including the most classic types, or those cited in the literature, of mutations that generate resistance in the target, as well as including the types of target (proteins, receptors or correlates) most susceptible to resistance and why. If it gets too long, consider creating a separate topic before “2. Non-systematic, preclinical anticipation of drug resistance mechanisms”. Details were added in lines 93-97
Lines 92 – 95: “Another non-systematic technology is to use lentiviral transduction...” if it's another technique found in the literature, then why hasn't it been covered with the same prominence? “2.1. Random mutagenesis”? It would be interesting to create a subtopic within “2. Non-systematic, preclinical anticipation of drug resistance mechanisms” addressing it in more depth.
If it is not well supported in the literature, then it is worth mentioning in the text the lack of papers dealing with it.
This is a very good point. The way it was written could have led to the confusion that a different, independent method exists. We have corrected this section, and hope it is now clear that the mentioned technology (Yenerall 2021) provides another opportunity for generation of mutant libraries, yet it still is an approach to investigate resistance mutations using random mutagenesis concept.
Lines 96 – 102: Are the statements regarding cost-benefit, use, and disadvantages reported in the researched literature or is it a conclusion of the authors? Please add references that address and emphasize these statements. The statement was reformulated and now says that random mutagenesis is more straightforward and does not need the design of dedicated libraries. Additional references were added concerning disadvantages of random mutagenesis.
2.2. Chronic exposure of tumor models to drug regimens
Lines 110 – 113: Add references that confirm the statements. Additional references were added
When citing a tissue-specific cancer (e.g. breast cancer), please also cite the lineage used in the study to which the cancer belongs (e.g. MCF-7, 4T1, MDA-MB-231). Please consider this suggestion for the entire manuscript. Details were added. Please note that in several cases, PDX models described in the literature do not have individual names.
Lines 161 – 163: Add a reference to support your statement. Please consider this suggestion for the entire manuscript. Done
- SYSTEMATIC, PRECLINICAL ANTICIPATION OF ON-TARGET DRUG RESISTANCE MECHANISMS
3.1. Deep Mutational Scanning (DMS): Ok
3.2. CRISPR-based base-editing screens: Ok
3.3. Computational methods: Ok
- SYSTEMATIC, PRECLINICAL ANTICIPATION OF OFF-TARGET DRUG RESISTANCE MECHANISMS
4.1. CRISPR-Cas9 knockout screens to identify genes and pathways that confer drug resistance
Line 333: It is interesting to modify "in recent years" by a specific and updated time range (e.g.: from 2022, CRISPRko has led...). Consider this suggestion for the entire manuscript and for other terms that refer to current dates without a defined period. Done
Line 380: “and inhibition of either pathway led to increased anti-tumor efficacy...”, how many % effective? It would be interesting to add the quantitative values of the findings to enhance the results. Consider this suggestion for other excerpts extolling better anti-tumor results. The reported in vivo efficacy data show similar efficacy, % inhibition values are howerver not given.
Lines 387 – 390: “of the enhancer of zeste homolog 2 (EZH2) inhibitor GSK126 revealed that the H3K36me2 methyltransferase NSD2 was connected with resistance (135).”, for this and the other passages throughout the manuscript, which expose possible mechanisms of resistance or not, it is interesting to delve deeper into the approach of the results. Did the studies cited compare tumors/grafts? Did they carry out other tests involving the mechanisms? If so, when possible, add the comparative values that the authors found with the control groups during the pre-clinical phase. Only in vitro data were generated in this paper. We added a sentence to explain the mechanistic impact of the methyltransferase on the SWI/SNF and PRC1/2 complexes.
4.2. CRISPR interference (CRISPRi) and CRISPR activation (CRISPRa) screens to reveal gene 418 expression modulations that confer drug resistance
Lines 460 – 663: “There were also findings specific to the method used, indicating that some phenotypes can only be observed under given experimental conditions” cite an example of a phenotype and condition. Is the reason for this event discussed, clarified or suggested? If so, I think it would be interesting to add it too. We have now further elaborated on the findings reported for the CRISPRko, CRISPRi and CRISPRa screens in this paper.
- DISCUSSION
It would be interesting to emphasize which of the techniques discussed in the manuscript proved to be the most promising, as well as to provide more details on their future prospects for widespread use. Due to complementarity, CRISPRi combined with CRISPRa represent the most promising approaches. This was added. More widespread use would be applications in personalized therapies, and this is already happening for hematological tumors (ref. 198 and 205).
Is the most promising technique the one most covered in the literature? If yes, is there a prevalence of studies addressing its pre-clinical use for which type of cancer/xenograft? If not, what prevents its further exploration to determine on- and off-target resistance mechanisms? Is it just the reasons highlighted at the end of the paragraphs of the reviews preceding the discussion? Many thanks for these suggestions. Xenograft models have mainly been used fro chronic exposure approaches (see paragraph 2.2).We have covered the CRISPR technology very extensively, as it is the most promsing but examples in xenografts are rarer. We have added and discussed several references in paragraph.
CONCLUSION
Even if the logical reasoning has been closed in the discussion, there needs to be a conclusion highlighting the nature of the manuscript, the main findings, the most promising techniques and, among them, the technique that showed the best pre-clinical performance for further exploration in future pre-clinical studies. We have added these points at the end.
Reviewer 2 Report
Comments and Suggestions for Authors
This is an interesting review that discusses drug resistance in cancer. Specifically, it highlights emerging strategies that use high-throughput screening technologies, such as the CRISPR system, in collaboration with computational methods to anticipate resistance mechanisms before they occur in treated cancer patients.
As the "on and off" target during cancer treatment is an important topic, a detailed discussion on this method is needed. I thus support its publication. However, some simple but important modifications can further important aspects of this review by providing an overall landscape of this new field of cancer drug resistance.
The concepts of cancer evolution need to be briefly discussed in this paper as they represent new frontiers in cancer drug-resistant research. As cancer evolution is driven by genome instability through a two-phased evolution, then aided by gene mutations in the second phase of microevolution, understanding the mechanisms of diverse gene mutation-mediated drug resistance requires a systemic perspective (see PMID: 33189848). For example, treatment-induced rapid drug resistance becomes a hot topic as treatments can induce genome chaos, including chromothripsis and PGCCs (Giant polyploid cancer cells) (PMID: 36596845; PMID: 34195196), which are mainly responsible for the "on and off" effect, as different genome systems alter the function of genetic networks.
Additionally, it should be mentioned that using the CRISPR platform can also alter the overall genome (outside the target) (PMID: 37803452; PMID: 36048170), which could paradoxically lead to drug resistance.
Comments on the Quality of English LanguageThe English is fine.
Author Response
Many thanks for reviewing our paper. Here how we addressed your points:
Review 2
This is an interesting review that discusses drug resistance in cancer. Specifically, it highlights emerging strategies that use high-throughput screening technologies, such as the CRISPR system, in collaboration with computational methods to anticipate resistance mechanisms before they occur in treated cancer patients.
As the "on and off" target during cancer treatment is an important topic, a detailed discussion on this method is needed. I thus support its publication. However, some simple but important modifications can further important aspects of this review by providing an overall landscape of this new field of cancer drug resistance.
The concepts of cancer evolution need to be briefly discussed in this paper as they represent new frontiers in cancer drug-resistant research. As cancer evolution is driven by genome instability through a two-phased evolution, then aided by gene mutations in the second phase of microevolution, understanding the mechanisms of diverse gene mutation-mediated drug resistance requires a systemic perspective (see PMID: 33189848). For example, treatment-induced rapid drug resistance becomes a hot topic as treatments can induce genome chaos, including chromothripsis and PGCCs (Giant polyploid cancer cells) (PMID: 36596845; PMID: 34195196), which are mainly responsible for the "on and off" effect, as different genome systems alter the function of genetic networks. These important points were now added in the introduction and in paragraph 2.2.
Additionally, it should be mentioned that using the CRISPR platform can also alter the overall genome (outside the target) (PMID: 37803452; PMID: 36048170), which could paradoxically lead to drug resistance. Many thanks. This is an important point, especially for the CRISPR ko approach and we have now added and discussed these 2 references in paragraph 4.1.
Reviewer 3 Report
Comments and Suggestions for Authors
The authors expose the literature correctly, is a a good revision.
Just some minor references missing:
- For global/international cancer statistics the authors should also include the following reference: CA Cancer J Clin. 2021 May;71(3):209-249. doi: 10.3322/caac.21660. Epub 2021 Feb 4.
- Line 96, reference missing at the endo of the sentence that defines random mutagenesis
- Line 129, not clear what classical inhibitors the authors are referring
- Line 199, reference should be added
- Line 345 references to the 2 2013 studies should be added here
Additionally, since the authors mostly focus on CRISPR technology, the tittle should reflect this.
Author Response
Many thanks for your suggestions. Here how we addressed your points:
Review 3
The authors expose the literature correctly, is a a good revision.
Just some minor references missing:
For global/international cancer statistics the authors should also include the following reference: CA Cancer J Clin. 2021 May;71(3):209-249. doi: 10.3322/caac.21660. Epub 2021 Feb 4. This reference was added and discussed in the Introduction.
Line 96, reference missing at the endo of the sentence that defines random mutagenesis. References were added
Line 129, not clear what classical inhibitors the authors are referring. This is now better explained.
Line 199, reference should be added. Done
Line 345 references to the 2 2013 studies should be added here Done
Additionally, since the authors mostly focus on CRISPR technology, the tittle should reflect this. As we also give extensive details on other strategies beyond CRISPR such as Deep Mutational Scanning, Random Mutagenesis, generation of resistant cell clones by long-term drug exposure, computational approahces, we would prefer to keep the title broader